# Metagenomics uncovers dietary adaptations for chitin digestion in the gut microbiota of convergent myrmecophagous mammals

Sophie Teullet,[1] Marie-Ka Tilak,[1] Amandine Magdeleine,[1] Roxane Schaub,[2,3] Nora M. Weyer,[4] Wendy Panaino,[4,5] Andrea Fuller,[4] W. J. Loughry,[6] Nico L. Avenant,[7] Benoit de Thoisy,[8,9] Guillaume Borrel,[10] Frédéric Delsuc[1]

**ABSTRACT**   In mammals, myrmecophagy (ant and termite consumption) represents a striking example of dietary convergence. This trait evolved independently at least five times in placentals with myrmecophagous species comprising aardvarks, anteaters, some armadillos, pangolins, and aardwolves. The gut microbiome plays an important role in dietary adaptation, and previous analyses of 16S rRNA metabarcoding data have revealed convergence in the composition of the gut microbiota among some myrmecophagous species. However, the functions performed by these gut bacterial symbionts and their potential role in the digestion of prey chitinous exoskeletons remain open questions. Using long- and short-read sequencing of fecal samples, we generated 29 gut metagenomes from nine myrmecophagous and closely related insectivorous species sampled in French Guiana, South Africa, and the United States. From these, we reconstructed 314 high-quality bacterial genome bins of which 132 carried chitinase genes, highlighting their potential role in insect prey digestion. These chitinolytic bacteria belonged mainly to the family Lachnospiraceae, and some were likely convergently recruited in the different myrmecophagous species as they were detected in several host orders (i.e., *Enterococcus faecalis*, *Blautia* sp.), suggesting that they could be directly involved in the adaptation to myrmecophagy. Others were found to be more host-specific, possibly reflecting phylogenetic constraints and environmental influences. Overall, our results highlight the potential role of the gut microbiome in chitin digestion in myrmecophagous mammals and provide the basis for future comparative studies performed at the mammalian scale to further unravel the mechanisms underlying the convergent adaptation to myrmecophagy.

**IMPORTANCE**   Myrmecophagous mammals are specialized in the consumption of ants and/or termites. They do not share a direct common ancestor and evolved convergently in five distinct placental orders raising questions about the underlying adaptive mechanisms involved and the relative contribution of natural selection and phylogenetic constraints. Understanding how these species digest their prey can help answer these questions. More specifically, the role of their gut microbial symbionts in the digestion of the insect chitinous exoskeleton has not been investigated in all myrmecophagous orders. We generated 29 new gut metagenomes from nine myrmecophagous species to reconstruct more than 300 bacterial genomes in which we identified chitin-degrading enzymes. Studying the distribution of these chitinolytic bacteria among hosts revealed both shared and specific bacteria between ant-eating species. Overall, our results highlight the potential role of gut symbionts in the convergent dietary adaptation of myrmecophagous mammals and the evolutionary mechanisms shaping their gut microbiota.

**KEYWORDS**   convergent evolution, myrmecophagy, mammals, gut microbiota, chitinases, metagenomics, genome assembly

Address correspondence to Sophie Teullet, sophie.teullet@umontpellier.fr, or Frédéric Delsuc, frederic.delsuc@umontpellier.fr.

The authors declare no conflict of interest.

See the funding table on p. 20.

In mammals, the gut microbiota has played a major role in dietary diversification, enabling transitions to novel carbon sources (1, 2). Several factors such as host diet, physiology, genetics, and phylogeny, but also environmental factors, shape the taxonomic composition and functional structure of the gut microbiota (3). Many studies have focused on convergent dietary adaptations and the effects of host diet, investigating how host phylogeny shapes the gut microbiota (4–10). A striking example of convergent dietary specialization is the adaptation to myrmecophagy in placental mammals. Myrmecophagous species are characterized by a diet composed of at least 90% of ants and/or termites (11). A total of 22 placental species evolved toward this diet and are found in five independent orders: Tubulidentata (aardvarks), Pilosa (anteaters), Cingulata (tolypeutine armadillos), Pholidota (pangolins), and Carnivora (aardwolves) (12–14). Their divergent evolutionary histories raise the question of how these species convergently adapted to the myrmecophagous diet and whether similar adaptive mechanisms were involved between the different species. Myrmecophagous species need to digest the chitinous exoskeleton of their prey to extract nutrients. Mammals carry chitinase genes in their genomes, which participate in chitin digestion (15–18), but their gut microbiota might also have played an important role in facilitating prey digestion (19–21).

Indeed, chitinolytic bacteria are present in a diversity of environments (i.e., soils, sediments, and aquatic environments) where they ensure chitin degradation and play an important role in the carbon cycle (22–25). They also have been identified in the digestive tract of mammals where they could participate in prey digestion (26–30). Taxonomic analyses based on 16S rRNA sequences have highlighted similarities in the gut microbiota composition of myrmecophagous species (19) compared to their non-myrmecophagous sister species. For instance, genera such as *Blautia* (Lachnospiraceae), *Streptococcus* (Streptococcaceae), *Peptococcus* (Peptococcaceae), or *Eubacterium* (Lachnospiraceae) were found to be significantly more abundant in the gut microbiota of myrmecophagous species than in their sister species. Focusing on the gut archaeome of placentals, *Methanobrevibacter* has been found almost only in myrmecophagous species (10). Yet, little is known regarding the role these symbionts play in the digestion of the chitinous exoskeleton of their prey.

Microbial chitinases and *N*-acetylglucosaminidases are carbohydrate-active enzymes (CAZymes) (31), which ensure the degradation of chitin into chitosan and are mostly found in the glycosyl hydrolases (GH) families 18, 19, and 20 (23, 32). Since bacterial chitinases mainly belong to the GH18 family (32–35), they represent relevant candidates to assess the potential of the gut microbiota to digest chitin in our focal species. The GH18 enzyme family comprises chitinases and chitin-binding proteins and within this family, bacterial chitinases are classified into three subfamilies (A, B, and C) based on sequence homology (32–34). Symbiotic chitin-degrading bacteria have been identified, as well as their chitinase genes, in the Malayan pangolin and the giant anteater (20, 21) and in other mammals having a chitin-rich diet (36, 37). This suggests that other myrmecophagous species might also carry chitinolytic gut bacteria that could participate in prey digestion. In the specific example of dietary convergence, one question resides in understanding whether these potential microbial mechanisms of chitin exoskeleton digestion have converged among the five different placental orders of myrmecophagous mammals. The different myrmecophagous mammal lineages diverged millions of years ago and evolved in diverse environments. Chitinolytic microbes might thus have been independently recruited to ensure chitin digestion, but whether similar bacteria carrying similar functions are involved still needs further investigation. Assessing whether chitinolytic bacteria present in myrmecophagous mammals are taxonomically and functionally similar will shed light on their origin.

To investigate the chitin-degrading potential of the gut microbiota of myrmecophagous mammals, we took advantage of recent advances in metagenomics to reconstruct high-quality bacterial genomes and identify GH18 as this enzyme family comprises most bacterial chitinases. By combining Nanopore long-read and Illumina short-read shotgun

metagenomic sequencing, we generated 29 new gut metagenomes from field-collected fecal samples of nine myrmecophagous and insectivorous species representatives of the five myrmecophagous placental orders. From these, we reconstructed 314 high-quality bacterial genomes harboring a diversity of GH18 chitinases. Identification of GH18 sequences in the same bacterial genomes revealed a potential role of gut symbionts in prey digestion through putative complex metabolic pathways. Both generalist and host-specific bacteria were identified in the different myrmecophagous species, potentially reflecting the divergent evolutionary histories of their hosts, and raising questions about the evolutionary forces (historical contingency and determinism) at play in shaping the gut microbiome of convergently evolved myrmecophagous mammals.

## RESULTS

### Potential role of the gut microbiota in chitin digestion

Combining long-read and short-read assemblies, we were able to reconstruct 314 dereplicated high-quality genome bins (156 and 158 from each dataset, respectively; see Table S1F) and highlighted the usefulness of using the two methods (see supplementary results part 1 available via Zenodo). The following analyses were done on this set of high-quality selected bins to assess the presence of chitinolytic bacteria in the gut microbiota of myrmecophagous mammals.

All selected bins were taxonomically assigned to bacterial genomes and their phylogeny is presented in Fig. S1. Overall, 58.3% of the bins ($n = 183$) were not assigned at the species level, 29.3% ($n = 92$) at the genus level, and 6.7% ($n = 21$) at the family level. Taxonomically assigned bins belonged mainly to the Lachnospiraceae ($n = 63$), Burkholderiaceae ($n = 24$), Acutalibacteraceae ($n = 18$), Ruminococcaceae ($n = 12$), and Bacteroidaceae ($n = 11$) families (Table S1F). The 314 selected bins were integrated into a phylogeny of prokaryote reference genomes to confirm these results (Fig. 1A). All selected bins were well placed in this phylogeny, suggesting they are similar to known bacterial genomes and not distantly related, even for genome bins for which taxonomic assignment failed. Some of the reconstructed bacterial genome bins clustered together within the Firmicutes (Fig. 1B), Bacteroidetes, and Proteobacteria (Fig. S2A and B), in clades not including reference genomes, suggesting that these genome bins could be specific to myrmecophagous species (here called myrmecophagous-specific clades). Within the Firmicutes, six clades containing more than two bins reconstructed from myrmecophagous samples can be defined within the Lachnospiraceae family with two clades including, respectively, 12 and 11 genome bins (Fig. 1B). In the first clade, six bins had no taxonomic assignment below the family level and could represent bacterial genomes not yet described, two belonged to the *Acetatifactor* genus, and four to the *CAG-510* genus. In the other clade, four bins were not taxonomically assigned below the family level, three bins belonged to the *CAG-590* genus, one was assigned to the *CAG-127* genus, and two were assigned at the species level (*Frisingiococcus caecimuris* and *Acetivibrio ethanolgignens*) (Fig. 1B). These bins were all reconstructed from xenarthran gut metagenomes (Fig. 1B). Within the Acutalibacteraceae family, one clade contained 18 genome bins reconstructed mostly from xenarthran gut metagenomes and the three aardvark samples (Fig. 1B). Eleven had no taxonomic assignment below the family level, four belonged to the *Eubacterium* genus, and three to the *UBA1227*, *UBA1691*, and *UBA6857* genera (Fig. 1B). Other clades containing only genome bins from myrmecophagous species were found within the Ruminococcaceae, Oscillospiraceae, Anaerovoracaceae, and Erysipelotrichaceae families.

To understand whether these bacteria could potentially degrade chitin, these high-quality genome bins were then used to search for GH18 enzymes. This resulted in 132 bins containing at least one GH18 sequence (between 0 and 17 bins per sample, 4.93 on average). Having complete genome bins enabled us to identify several GH18 genes in the same bin (between 0 and 17 per bin, 1.26 on average; Table S1F). In total, 394 GH18 sequences were identified (Fig. 2) with 237 sequences presenting an active chitinolytic site (DXXDXDXE) (Fig. 2) and distributed among 82 bins (here called chitinolytic bins).

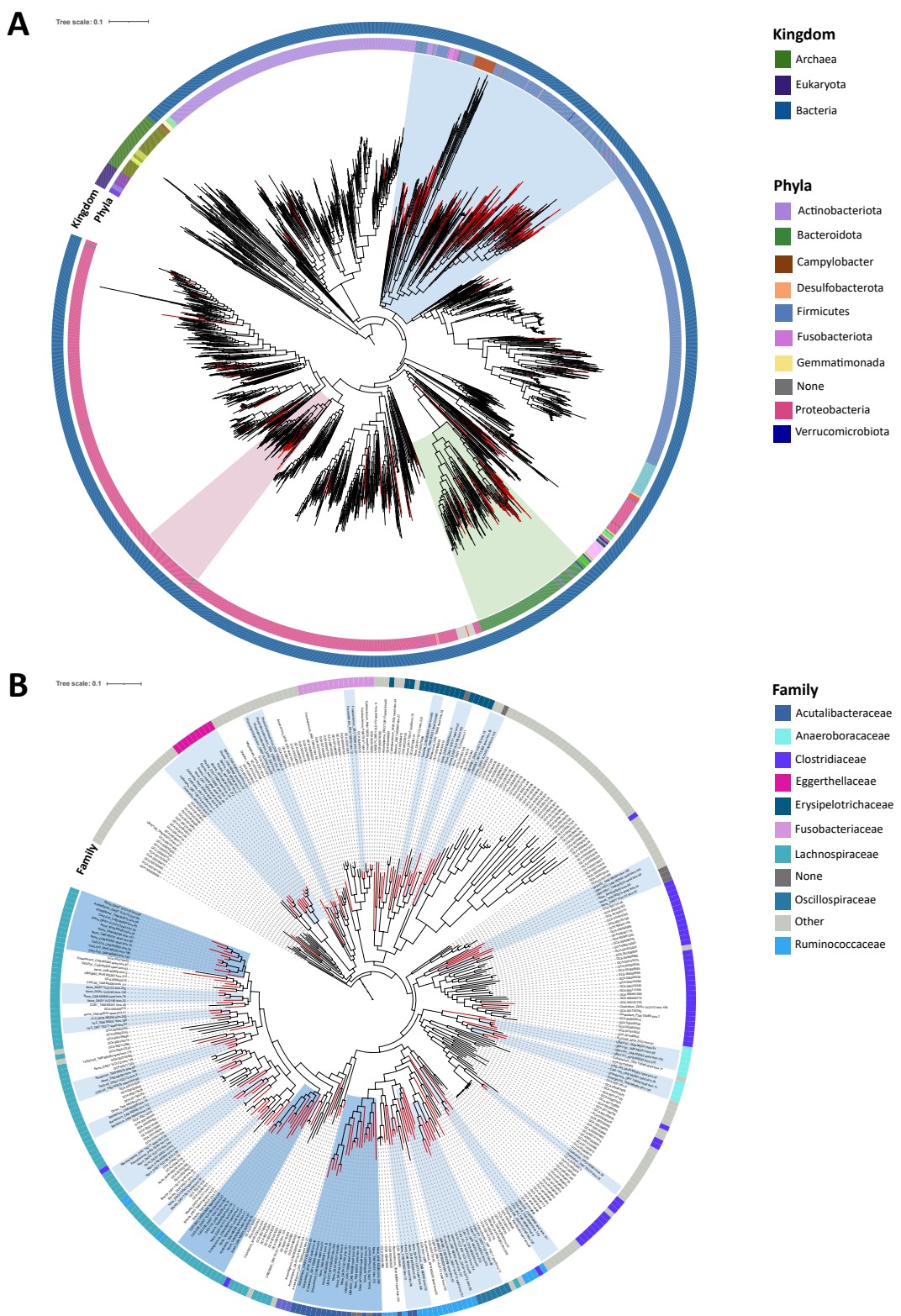

**FIG 1** Phylogenetic position of the 314 high-quality selected bins reconstructed from 29 gut metagenomes of the nine focal myrmecophagous species within a reference prokaryotic phylogeny. (A) Phylogeny of the 314 selected bins (red branches) with 2496 prokaryote reference genomes. Circles, respectively, indicate (from inner to outer circles): the bacterial phyla and kingdom to which these genome bins were assigned based on the Genome Taxonomy Database release 7 (Continued on next page)

**FIG 1** (Continued)

(38). Clades, where a subtree was defined, are highlighted in blue for the Firmicutes (Fig. 1B), green for the Bacteroidetes, and pink for the Proteobacteria (Fig. S2A and B, respectively). (B) Subtree within Fimircutes showing myrmecophagous-specific clades (blue highlights; dark blue corresponds to the three clades mentioned in the results, light blue to the other clades). The outer circle indicates the bacterial family to which these genome bins were assigned based on the Genome Taxonomy Database. Bins' names of the myrmecophagous-specific clades are indicated at leaves of the phylogenetic tree together with the genus to which they were assigned to.

These chitinolytic sequences are found in genome bins belonging mainly to the Lachnospiraceae (*n* = 183 sequences; e.g., *Blautia*, *Acetatifactor*, *Roseburia*, *Clostridium* genera), Acutalibacteraceae (*n* = 76; e.g., *Eubacterium* genus), and Ruminoccocaceae (*n* = 23; e.g., *Ruminococcus*, *Acetanaerobacterium* genera) bacterial families (Fig. 2). Fifty-three sequences not presenting an active site were found in a clade with sequences similar to lysin motif (LysM) domain-containing proteins (Fig. 2), which is a 40 amino acid domain involved in peptidoglycan and chitin-binding (39). Sixty-two were placed in a clade with sequences similar to src Homology-3 (SH3) domain-containing proteins (Fig. 2), which is a 50 amino acid domain found in intracellular and membrane proteins involved in the binding of ligands (40). These two clades were used to root the tree. Finally, 278 GH18 sequences formed a clade with sequences similar to known bacterial chitinases (Fig. 2). The majority had an active chitinolytic site. These sequences were identified in bins reconstructed from different host species, representative of the five myrmecophagous placental orders. They represent a diversity of chitinase genes, as they are distributed in distinct clades (Fig. 2).

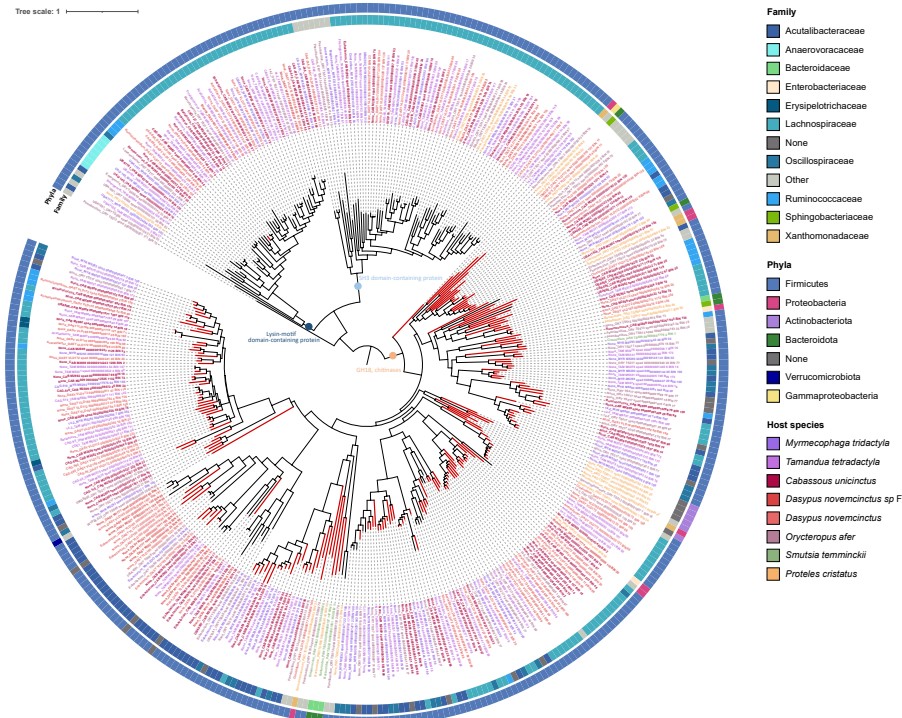

**FIG 2** Phylogeny of the 394 GH18 sequences identified in 132 high-quality selected bins reconstructed from 29 gut metagenomes of the nine focal myrmecophagous species and relatives. Red branches indicate the 237 sequences having an active chitinolytic site (DXXDXDXE). Circles, respectively, indicate (from inner to outer circles): the bacterial family and phyla of the bin the sequence was retrieved from. Colored sequence names indicate the host species. Colored circles at certain nodes indicate enzymes to which sequences are similar when blasting them against the NCBI nonredundant protein database. Sequence names are indicated at leaves of the tree and begin with the genus to which the bin they were identified in was assigned to.

## Distribution of chitinolytic bacteria among myrmecophagous mammals

Numbers of shared and specific selected bins in the nine host species were computed based on a detection threshold of 0.25 (the percentage of the reference covered by at least one read) of the selected bins across samples (Fig. 3; Table S2 , and detection table available via Zenodo). According to this threshold, five selected bins were not considered to be detected in any sample. Between 6 and 124 selected bins were detected in each sample (52.62 on average) and 252 selected genomes were shared and detected in samples other than the ones they were reconstructed from (Fig. 3). Selected bins were detected in 4.86 different samples on average (between 0 and 18). Selected bins were shared between individuals of the same species but with some intra-specific variability (Fig. 3). Conversely, 57 bacterial genomes were detected in only one sample (Fig. 3; Table S2) and 194 bins were detected in more than one host species (between 0 and 7 host species; 2.73 on average), between closely related species or distantly related species (Fig. 3; Table S2). On the other hand, 115 selected bins were specific to a particular host

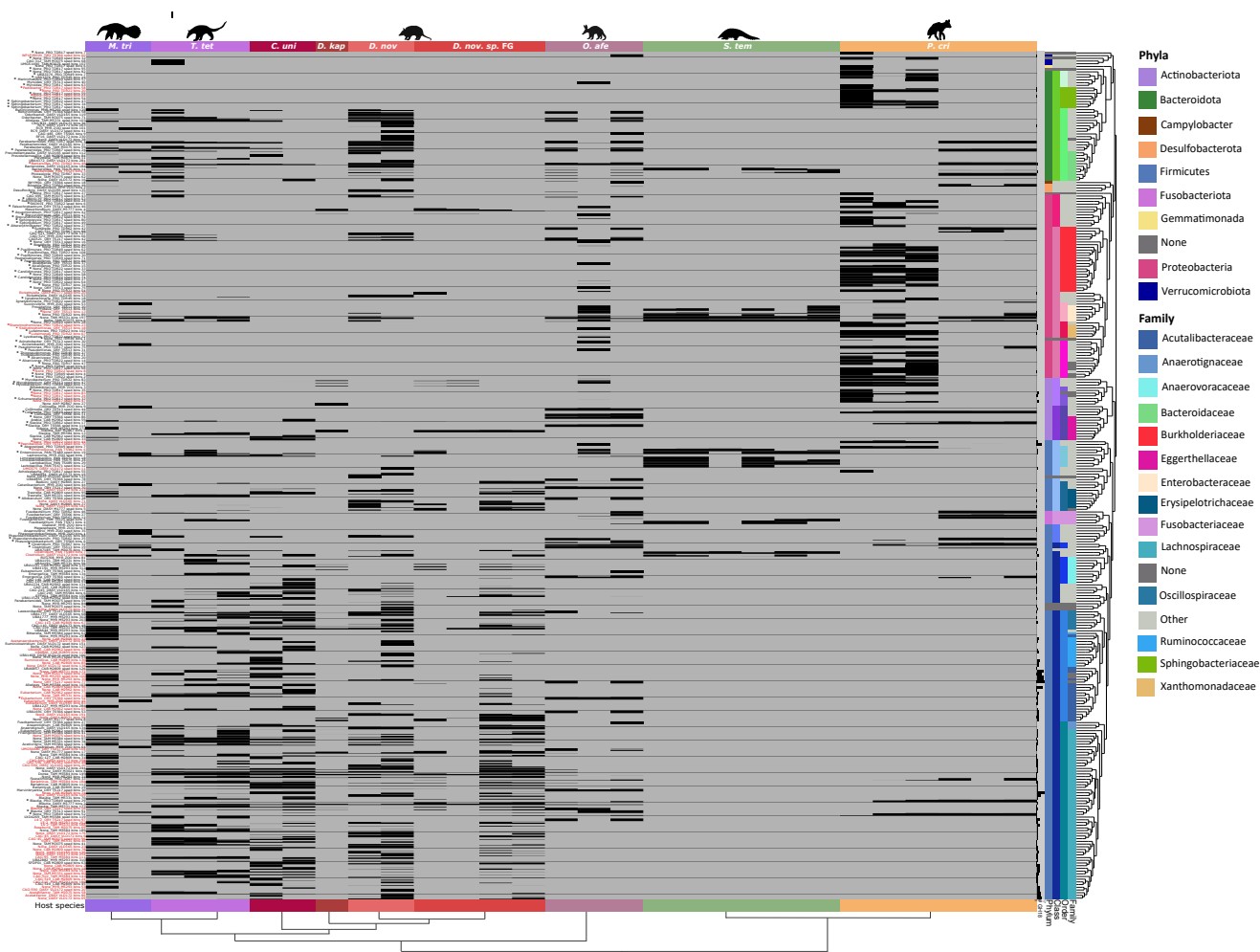

**FIG 3** Detection of the 314 high-quality bacterial genomes (lines) in the 29 gut metagenomes (columns) of the nine focal species. Each square indicates the detection of a genome bin in a sample as estimated by anvi'o v7 (41). Names of bins are indicated on the left with red indicating chitinolytic bins (Table S2). The names begin with the genus to which the bin was assigned to. Asterisks (*) indicate bins detected in at least one soil sample (detection >0.25) (Fig. S4; Table S2, and detection table available via Zenodo). Phylogenetic relationships of host species distinguished by different color strips are represented at the bottom of the graph. Columns on the right indicate (from left to right) the number of GH18 sequences identified in each bin (from 0 to 17), the bin's taxonomic phylum, class, order, and family. The phylogeny of the 314 selected bins inferred with PhyloPhlAn v3.0.58 (42) is also represented on the right of the graph (see Fig. S1). Silhouettes were downloaded from phylopic.org.

species, meaning they were found in only one of the nine species studied here (Fig. 3; Table S2). Selected bins were detected in 1.80 different host orders on average (between 0 and 5) for a total of 179 bins detected in more than one host order. Finally, 130 bins were detected in only one host order.

There were more shared selected bins between host species carrying GH18 genes than shared bins with no GH18 gene (Fig. S3). Among the 194 bins shared between host species, 110 had at least one GH18 gene. These bins belonged to 15 different bacterial families, mainly the Lachnospiraceae (*n* = 60; e.g., *Blautia*, *Acetatifactor*, *Roseburia* genera) and Acutalibacteraceae (*n* = 10; e.g., *Eubacterium* genus) (Fig. S3). The 84 shared selected bins without chitinases belonged to a more diverse range of bacterial families (*n* = 33), mainly Burkholderiaceae (*n* = 8; e.g., *CAG-521*, *Sutterella*, *Bordetella* genera) and Bacteroidaceae (*n* = 7; e.g., *Bacteroidetes*, *Prevotella* genera) (Fig. S3). On the contrary, among bins found in only one host species, there were fewer chitinase-carrying bins. Among the 115 host-species-specific bins, 21 had at least one GH18 gene. They were found in 12 different bacterial families (Fig. S3). The 94 specific bins with no chitinase genes belonged to 37 different bacterial families, mainly Burkholderiaceae (*n* = 16) (Fig. S3). A similar pattern was observed at the host order level (see supplementary results part 2 available via Zenodo).

Three high-quality selected bins carrying GH18 were shared between species belonging to the five different myrmecophagous orders (Table S1F): one bin of *Bacteroides fragilis* (mean absolute abundance across samples 2.39, see abundance table available via Zenodo) and two bins of Enterobacteriaceae (mean absolute abundances across samples 0.58 and 3.66). Two of these three bins shared across orders were a bit more abundant, on average, than the mean absolute abundance of selected bins across samples of 0.89 (the minimal abundance of a bin across samples was 2.82e-6 and the maximal absolute abundance was 198). Seven selected bins carrying GH18 were shared among four myrmecophagous orders (Table S1F). Two bins belonged to *Enteroccocus faecalis* and were reconstructed from pangolin samples, along with bins belonging to Lachnospiraceae bacteria (notably one from *Blautia* sp., one from *Faecalimonas* sp., and one from an unknown genus), one from *Bacteroides* sp. (Bacteroidaceae), and one from *Emergencia timonensis* (Anaerovoracaceae) with mean absolute abundances across samples ranging from 0.21 (*E. timonensis*) to 4.74 (*Bacteroides* sp.), the latter being found to be abundant in several samples [e.g., one *D. novemcinctus* (12.2), one *T. tetradactyla* (21.2), one *O. afer* (41.2)]. One bin of *Enteroccus faecalis* also had a mean absolute abundance (3.43) above the mean absolute abundance of selected bins and was found to be abundant in several samples as well [one *D. sp. nov* FG (33.8), three pangolin samples (10.8, 13.3 and 21.8), and one *T. tetradactyla* (15.3)].

At least one chitinolytic bin (a high-quality selected bin carrying at least one GH18 sequence with an active chitinolytic site) was detected in the gut microbiota of each host species (Fig. 4; Table S2). Consistent with our previous results (Fig. 2) these bins mainly belonged to the Firmicutes (Fig. 4). More chitinolytic bins were detected in gut metagenomes of xenarthran species compared to the aardvark, the southern aardwolf, and the ground pangolin, except for *Dasypus kappleri* for which we only had one sample (Fig. 4; Table S3). When compared to the total number of genome bins detected per species, xenarthran species had higher proportions of chitinolytic genome bins (Table S3).

## DISCUSSION

### Potential role of the gut microbiota in prey digestion in myrmecophagous mammals

The gut microbiota plays an important role in host digestion, which has led to several cases of gut microbiome convergence in distantly related species that share similar diets (5, 19, 44). In animals with a chitin-rich diet, symbiotic chitinolytic bacteria could be involved in prey digestion, as they have been identified in species as diverse as nine-banded armadillos (45), insectivorous bats (28), insectivorous monkeys (30), or

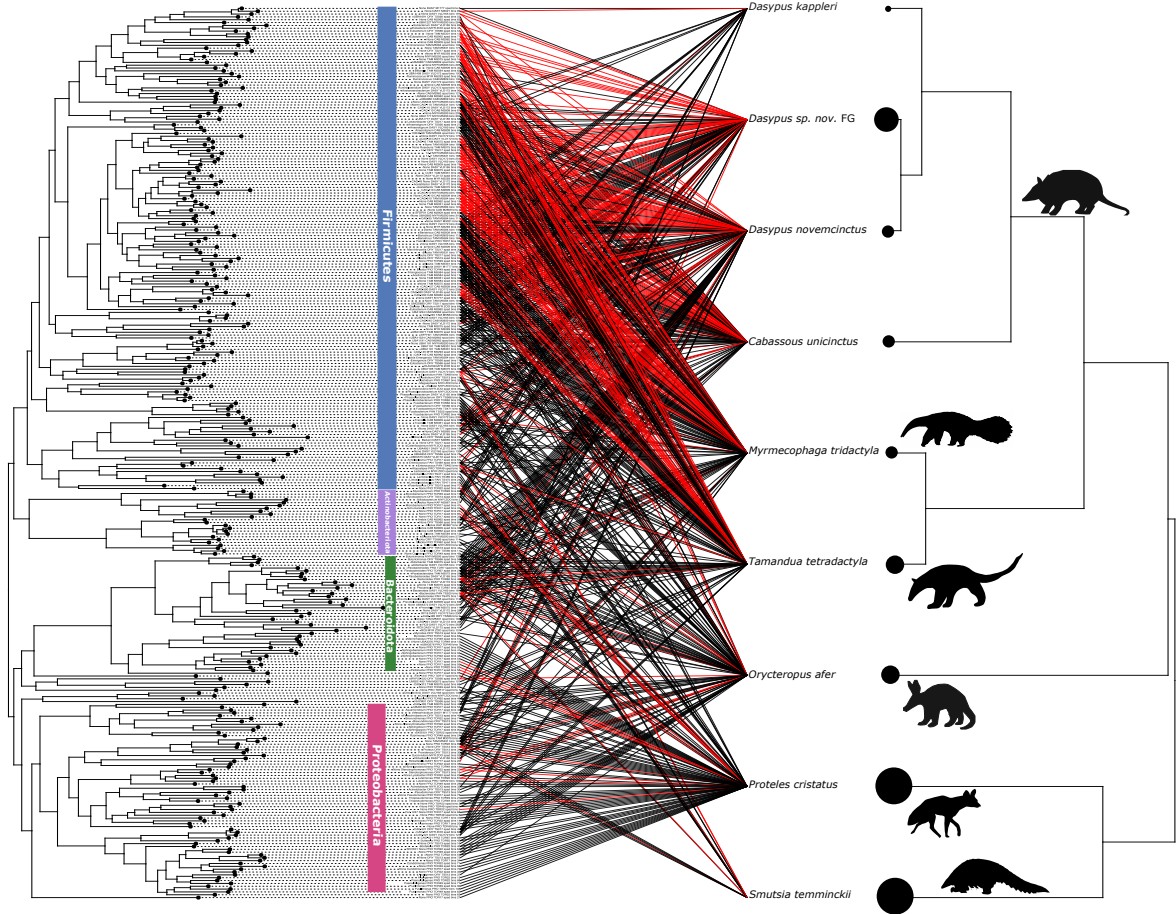

**FIG 4** Distribution of chitinolytic selected bins (red links) among the nine focal myrmecophagous species and relatives. Phylogenies of the 314 high-quality selected bins (Fig. S1) and of the nine host species (downloaded from timetree.org) are represented, respectively, on the left and the right of the graph. Links illustrate, for each bin, in which host species the bin was detected (detection threshold >0.25). Red links indicate bins in which at least one GH18 sequence with an active chitinolytic site (DXXDXDXE) was found (chitinolytic bins). The size of the circles at the tips of the host phylogeny is proportional to the number of samples (n = 1 for *D. kap*; n = 2 for *D. nov*, *C. uni* and *M. tri*; n = 3 for *T. tet* and *O. afe*; n = 4 for *D. sp. nov* FG; n = 6 for *P. cri* and *S. tem*). Bins' names are indicated at the tip of the bins' phylogeny and main bacterial phyla are indicated by colored vertical bars. This graph was done with the cophylo R package within the phytools suite (43). Silhouettes were downloaded from phylopic.org.

crustacean-eating whales (46). Here, we focused on reconstructing high-quality bacterial genome bins from gut metagenomes of species representative of the five myrmecophagous orders and identified sequences belonging to the main bacterial chitinases (i.e., GH18) (32, 33, 47). This allowed us to assess the potential of chitin degradation of gut symbionts in myrmecophagous placentals and better understand their convergent adaptation to this highly specialized diet.

Among the high-quality bacterial genome bins reconstructed, several clustered together in myrmecophagous-specific clades. These clades were found mainly within the Lachnospiraceae and Actualibacteraceae families within Firmicutes but also within the Oscillospiraceae and Ruminococcaceae. Several of these genomes had no taxonomic assignment below the family level, suggesting they could represent bacterial taxa not described yet and could be linked with the adaptation to myrmecophagy. Most of these bacterial taxa (i.e., Lachnospiraceae such as *Blautia* or *Roseburia* genera or Ruminococcaceae) were previously found to be significantly more abundant in the gut microbiota of the aardwolf, giant anteater, and southern tamandua when compared to

their non-myrmecophagous sister species based on 16S rRNA metabarcoding data (19), which suggests that their presence may be associated with dietary adaptations. Besides, chitin-degrading metabolic pathways were identified in the gut metagenome of the Malayan pangolin (*Manis javanica*) as well as genes encoding chitinases and chitin-binding proteins belonging to the GH18 and GH19 enzyme families (20). This allowed the identification of chitinolytic symbionts such as *Enterococcus faecalis* (Enterococcaceae), *Clostridium paraputrificum* (Clostridiaceae), and *Bacteroides fragilis* (Bacteroidaceae) (20), whose genomes were also reconstructed here from Temminck's pangolin (*Smutsia temminckii*) gut metagenomes, confirming their probable role in prey digestion in pangolins.

Numerous GH18 genes were identified within the high-quality bacterial genomes we reconstructed here. We were able to identify 132 genome bins carrying at least one GH18 gene among which 83 had at least one GH18 sequence with an active chitinolytic site (DXXDXDXE) (48, 49), here called chitinolytic bins, and representing bacteria that might thus play a role in insect prey digestion. Among genome bins found in the three myrmecophagous-specific clades mentioned previously, almost all had at least one GH18 sequence. This further suggests that these genome bins could represent bacterial taxa playing a significant role in chitin digestion in myrmecophagous species. Studying the distribution of these bacteria in other insectivorous mammals will allow further understanding of their role in the adaptation toward these specialized diets.

In addition, some of the bacterial taxa we identified here as being potentially involved in chitin digestion in myrmecophagous species belonged to or were closely related to bacterial taxa known for their chitinolytic properties. For example, *Clostridium* species are known for their chitinolytic activity (50–52). Chitinases have also been identified and studied in *Enterococcus faecalis* (53, 54). More generally, Lachnospiraceae, for which we reconstructed the most genomes, are known to degrade complex polysaccharides (55) and our results suggest they could be involved in chitin hydrolysis. Moreover, several genomes of Ruminococcaceae bacteria reconstructed here encode a GH18. Within this family, *Ruminococcus* species have been suggested to play a role in chitin digestion in the gut microbiota of insectivorous mammals (56). Several GH18 sequences were found in the same bacterial genomes, suggesting that complex chitin-degrading pathways are present in the gut microbiota of the myrmecophagous species, consistent with previous analyses of the chitinolytic properties of certain bacteria. Indeed, several enzymes are involved in the different steps of chitin hydrolysis (23). For example, two types of chitin-binding and chitinase enzymes have been described in *E. faecalis* (54). This bacteria also carries endo-β-*N*-acetylglucosaminidases (57) which are part of the chitin hydrolysis pathway and degrade chitin into chitosan (23). This further highlights the complex chitinolytic machinery of these bacteria.

Finally, some sequences do not have the ability to hydrolyze chitin because no active site was identified, but could still bind chitin. For example, the lysin motif (LysM) domain of certain proteins can bind peptidoglycan (39). It is present in certain eukaryotes' chitinases (i.e., algae, nematodes) and participates in the recognition of symbiotic rhizobial bacteria by leguminous plants and is thus thought to bind chitin (39, 58). The SH3 domain is also known to have binding properties and is involved in protein interactions (40). Therefore, sequences similar to LysM and SH3 domain-containing proteins could still be part of the chitin hydrolysis process by influencing molecular interactions. In addition, enzymes carrying the carbohydrate-binding module 37 (CBM37), which is known to have binding properties notably to chitin, could also be involved in the chitin-degrading process. The presence of other enzymes potentially involved in insect digestion could be investigated, such as trehalases, enzymes that break down trehalose, a sugar found in insect blood. The gut microbiota of the Malayan pangolin and giant anteater were found to be significantly enriched in these enzymes when compared to non-myrmecophagous species (21). The gut microbiota also plays a role in the detoxification of ingested compounds (26, 44), such as insect toxins.

For example, metabolic pathway analyses have revealed the potential of microbial symbionts to detoxify formic acid in some myrmecophagous species (21).

## Shared and species-specific gut bacteria among myrmecophagous mammals

Morphological and genomic comparative studies have shown that the convergent adaptation to myrmecophagy in mammals involved different mechanisms between the different species and that phylogenetic constraints played a central role (15, 18, 59–61). As diet is one of the main factors shaping the evolution of the gut microbiota (5), this raises the question of whether the same bacterial symbionts with similar functions were convergently recruited between the different ant-eating species, or whether it was different bacteria with similar functions (functional convergence).

Analysis of the distribution of selected genome bins across samples revealed both shared and host-specific bacteria among myrmecophagous species. Selected genomes were mainly shared between closely related species and mainly among *Xenarthra* but less so between distantly related orders such as pangolins (Pholidota) and anteaters (Pilosa), highlighting the influence of host evolutionary history. In mammals, phylo-symbiosis is particularly strong (9, 62) in part because of mammalian-specific traits such as viviparity or parental care (63) but also limited microbial dispersal abilities (64) facilitating the vertical transmission of microbes, which may explain these results. Some bacterial genomes were shared between distantly related host species and those shared across more than one order mainly belong to the family Lachnospiraceae. One selected genome of *Bacteroides fragilis* and two of Enterobacteriaceae were shared between species belonging to the five different myrmecophagous orders and carry GH18. Similarly, seven selected genome bins were shared among four host orders and belong to *Enteroccocus faecalis*, the Lachnospiraceae (i.e., *Blautia* sp., *Faecalimonas* sp.), *Bacteroides* sp. (Bacteroidaceae), and *Emergencia timonensis* (Anaerovoracaceae). These shared high-quality selected genome bins carrying GH18 may thus participate in the adaptation to myrmecophagy and highlight the influence of the host diet, consistent with previous studies showing that it is an important factor in shaping the mammalian gut microbiota (1, 4, 5, 8). Each host species appears to carry chitinolytic bacteria in its gut microbiota, confirming their potential role in adapting to an insect-based diet. However, differences in the distribution of these chitinolytic selected bins could highlight divergent microbiota adaptations, especially in xenarthran species, which carry more chitinolytic bacteria than aardvark, ground pangolin, and southern aardwolf. Some bins were found to be abundant in some samples raising questions on how the gut micro-biota is involved in prey digestion. Having few highly abundant chitinolytic bacterial species producing large amounts of chitinases might be sufficient for a specific host to digest its prey, whereas having diverse chitinolytic bacterial species might have been selected for in other host species.

The distribution of high-quality selected genome bins across samples and species revealed how shared and host-specific bacteria may reflect the divergent evolutionary histories of host species and the effect of adapting to a similar diet despite phylo-genetic constraints. This relates to the evolution of the mammalian gut microbiota. From a mammalian ancestor that probably had an insectivore-like gut microbiota (62), its composition might have changed as placentals diversified and occupied new niches, constraining its composition in certain species. Chitinolytic bacteria shared by all myrmecophagous orders may represent ancient bacterial lineages inherited from a common ancestor, while bacteria found in only certain host species may reflect more recent adaptations with bacteria acquired in specific lineages.

The environment and biogeography also influence host-associated microbial communities, which could explain the distribution pattern of selected bins across samples, with large differences observed between South Africa and South America. Anteaters and armadillos (*Xernarthra*) diverged anciently from aardvarks, pangolins, and aardwolves (~80 Ma) (14), and these lineages have evolved in very distinct environments and biogeographic contexts for most of their evolutionary history. Soil samples from the

fecal sampling sites have been collected for the South African samples. Some selected bins reconstructed from gut metagenomes of the ground pangolin, southern aardwolf, and aardvark belonged to bacterial taxa that were indeed not expected to be found in the gut but rather in the environment (e.g., Sphingobacteriaceae, Xanthomonadaceae, and Burkholdericaeae). The distribution of selected genome bins reconstructed from gut metagenomes of these species was examined in soil samples and showed that 91 selected bins were also detected in at least one soil sample (detection >0.25; Fig. S4; Table S2). For example, among the Proteobacteria selected genome bins found in myrmecophagous-specific clades (Fig. S2B), some belonged to Burkholderiaceae bacteria and were detected in at least one soil sample. This may not necessarily reflect environmental contamination, as the center of the feces was specifically sampled to minimize contamination. Rather, it may reflect the fact that these species ingest soil and thus environmental microbes while foraging (11). This could be beneficial to the host (65), as it may compensate for the lack of mastication and/or help to deal with toxins that might be present in prey (11). Environmental acquisition of microbes has also been suggested in lemurs (66) and wild echidnas (67), species that ingest soil during foraging and in which soil bacteria have been identified in their gut microbiota. These environmental bacteria could therefore represent transient bacteria in the gut, reflecting what is ingested by the host. They could also represent resident bacteria that have been recruited from the environment into the gut microbiota of myrmecophagous species. Indeed, some beneficial microorganisms could be acquired from soil microbiomes (68), which could allow the host to better adapt to its environment through horizontal gene transfer of beneficial genes (69). Moreover, it has also been suggested that environmental bacteria could be a source of chitinase genes to digest prey (22). Thus, their recruitment into the gut microbiota may have been selected to participate in the host digestion and further contribute to the adaptation to its environment. This would highlight the potential influence of the environment on the gut microbiota of myrmecophagous mammals.

## Role of the holobiont in prey digestion in myrmecophagous mammals

The influence of the microbiota on host evolution and their coevolution is increasingly recognized as studies of the diverse microbiota of captive and wild animals proliferate. Many scientists now recognize the term holobiont to describe the host and all its associated symbionts, which would be the unit of natural selection as defined by the hologenome theory of evolution (70). Integrative studies (21, 71) and initiatives such as the "Earth Hologenome Initiative" (http://www.earthhologenome.org/) to study hologenomic adaptations are now becoming more common. In the specific case of myrmecophagous mammals, a hologenomic approach would help to better understand the adaptive mechanisms involved. Indeed, if chitinolytic bacteria can digest chitin, prey digestion can also be ensured by host-produced chitinases. In mammals, from a placental ancestor that probably carried five functional chitinase paralogs (CHIA), some of these paralogs were subsequently lost during placental diversification in non-insectivorous species, leading to a positive correlation between the number of functional chitinase paralogs and the proportion of invertebrates in the diet (15); a correlation also observed in primates (72). Among myrmecophagous species, this chitinase gene repertoire has evolved differently to ensure chitin digestion, reflecting phylogenetic constraints (15, 18). Thus, both endogenous and microbial chitinases may be involved in prey digestion, raising the question of the relative contribution of the host and its symbionts in providing the same function. For example, myrmecophagous mammals with only one functional *CHIA* paralog (*CHIA5*), such as aardwolves and pangolins, may compensate for this by overexpressing their only functional paralog or by relying more on their chitinolytic symbionts to digest their prey. In pangolins, *CHIA5* has been found to be overexpressed in all digestive organs (18). In the aardwolf, which recently diverged from the other Hyaenidae species (~10 Ma) (73), and does not present strong morphological adaptations to myrmecophagy, no host chitinases have been found to be expressed in its salivary glands (18) and expression of these enzymes in other

digestive organs needs further investigation. Therefore, this species could rely more on its gut microbiota to ensure prey digestion. Based on our results, fewer chitinolytic bacteria were found in the gut microbiota of the southern aardwolf compared to other species but they could still ensure chitin digestion, for instance, by overexpressing their chitinases. The combination of the host genomic and transcriptomic data with metagenomic and metatranscriptomic data of its microbiota would allow answering this type of question by comparing host gene repertoires in light of the chitinolytic abilities of the gut microbiota. This will shed light on the adaptation to chitin digestion in placentals and, more generally, on the role of the holobiont in the adaptation to a specific function.

## MATERIALS AND METHODS

### Sampling

Thirty-three fecal samples were collected from nine species representative of the five myrmecophagous placental orders (Table 1). For armadillos and anteaters provided by the JAGUARS collection (Cayenne, French Guiana), fecal samples from roadkill and deceased zoo animals were obtained after unfreezing the specimens and dissecting the lower part of the digestive tract in the lab facilities provided by Institut Pasteur de la Guyane (Cayenne, French Guiana). Roadkill armadillos collected in the United States were also dissected to sample feces. For the aardvark, ground pangolin, and southern aardwolf, fresh fecal samples were collected directly in the field during fieldwork sessions conducted in Tswalu Kalahari and Tussen-die-Riviere reserves (South Africa). The inner part of the feces was sampled with a sterile scalpel blade to avoid soil contamination. In the South African reserves, eight soil samples were also collected near feces sampling sites to serve as a control for potential environmental contamination (Table S4). All fecal and soil samples were stored at −20°C in 96% ethanol before DNA extraction.

### DNA extraction

Whole DNA was extracted from fecal samples following the optimized protocol of (79) using the enomic DNA from soil kit (NucleoSpin, Macherey-Nagel). Two successive extractions were done and a purification step was added to retrieve high-molecular-weight DNA suitable for long-read sequencing (79). The same kit was used to extract DNA from soil samples. Before the extraction, samples were incubated with 700 µL of the lysis buffer (SL1) (79) and 30 µL of proteinase K at 56°C for 30 min.

### Library preparation and DNA sequencing

#### *Long-read sequencing*

Long-read libraries were constructed using the SKQ-LSK109 and SKQ-LSK110 library preparation kits (ONT, Oxford Nanopore Technologies, https://nanoporetech.com/). Shotgun metagenomic sequencing was done using the MinION, MK1C, and GridION devices using one R9 flowcell per sample. Between 150 and 988 ng of DNA were loaded per flowcell, which were run for 48 to 72 h (Table S1A). For samples sequenced on the GridION, super-accurate basecalling was performed with Guppy v5+ (Qscore = 10) whereas on the MinION and MK1C fast basecalling was used (Qscore = 7). Sequencing output statistics were checked using PycoQC v2.5.2 (80) with a minimum quality score set to seven or 10 depending on the sequencing device used (Table S1A). Rebasecalling of samples sequenced on the MinION and MK1C was done on a GPU machine with Guppy v5.0.16 (super accurate mode, Qscore = 10, config_file = dna_r9.4.1_450bps_sup.cfg) (ONT).

**TABLE 1** Detailed sample information for the 33 fecal samples collected[a,b]

| Sample name | Species name | Common name | Order | Diet | Sex | Age | Wild/Captive | Country of origin | Location | Accession number Illumina | Accession number ONT |
|---|---|---|---|---|---|---|---|---|---|---|---|
| CAB M2809 | Cabassous unicinctus | Southern naked-tailed armadillo | Cingulata | Myrmecophagous | Male | Adult | Wild | French Guiana | coll. JAGUARS | SRR23925023 | SRR23925022 |
| CAB M2962 | Cabassous unicinctus | Southern naked-tailed armadillo | Cingulata | Myrmecophagous | Male | Adult | Wild | French Guiana | coll. JAGUARS | SRR23925011 | SRR23925000 |
| CAB M3141 | Cabassous unicinctus | Southern naked-tailed armadillo | Cingulata | Myrmecophagous | Female | Adult | Wild | French Guiana | coll. JAGUARS | SRR23924989 | SRR23924978 |
| DASY M1746 | Dasypus novemcinctus sp FG | Guianan long-nosed armadillo | Cingulata | Omnivorous | Male | Adult | Wild | French Guiana | coll. JAGUARS | NA | SRR23924967 |
| DASY M1777 | Dasypus novemcinctus sp FG | Guianan long-nosed armadillo | Cingulata | Omnivorous | Male | Juvenile | Wild | French Guiana | coll. JAGUARS | SRR23924962 | SRR23924961 |
| DASY M2255 | Dasypus novemcinctus sp FG | Guianan long-nosed armadillo | Cingulata | Omnivorous | NA | Adult | Wild | French Guiana | coll. JAGUARS | SRR23924960 | SRR23925021 |
| DASY M2865 | Dasypus novemcinctus sp FG | Guianan long-nosed armadillo | Cingulata | Omnivorous | Female | Adult | Wild | French Guiana | coll. JAGUARS | SRR23925020 | SRR23925019 |
| DASY M3021 | Dasypus novemcinctus sp FG | Guianan long-nosed armadillo | Cingulata | Omnivorous | Male | Adult | Wild | French Guiana | coll. JAGUARS | SRR23925018 | SRR23925017 |
| DASY VLD165 | Dasypus novemcinctus | Nine-banded armadillo | Cingulata | Omnivorous | Female | Adult | Wild | USA | Valdosta (GA) | SRR23925016 | SRR23925015 |
| DASY VLD168 | Dasypus novemcinctus | Nine-banded armadillo | Cingulata | Omnivorous | Male | Adult | Wild | USA | Valdosta (GA) | NA | SRR23925014 |
| DASY VLD172 | Dasypus novemcinctus | Nine-banded armadillo | Cingulata | Omnivorous | Female | Adult | Wild | USA | Valdosta (GA) | SRR23925013 | SRR23925012 |
| KAP M2867 | Dasypus kappleri | Greater long-nosed armadillo | Cingulata | Insectivorous | Male | Adult | Wild | French Guiana | coll. JAGUARS | SRR23925010 | SRR23925009 |
| MYR M5293 | Myrmecophaga tridactyla | Giant anteater | Pilosa | Myrmecophagous | Female | Adult | Wild | French Guiana | coll. JAGUARS | SRR23925008 | SRR23925007 |
| MYR M5295 | Myrmecophaga tridactyla | Giant anteater | Pilosa | Myrmecophagous | NA | Juvenile | Wild | French Guiana | coll. JAGUARS | SRR23925006 | SRR23925005 |
| MYR ZOO | Myrmecophaga tridactyla | Giant anteater | Pilosa | Myrmecophagous | Female | Adulte | Captive | France | Montpellier zoo | SRR23925004 | SRR23925003 |

**TABLE 1** Detailed sample information for the 33 fecal samples collected[a,b] (Continued)

| Sample name | Species name | Common name | Order | Diet | Sex | Age | Wild/Captive | Country of origin | Location | Accession number | |
|---|---|---|---|---|---|---|---|---|---|---|---|
| ORY TS217 | Orycteropus afer | Aardvark | Tubulidentata | Myrmecophagous | NA | NA | Wild | South Africa | Tswalu Kalahari reserve | SRR23925002 | SRR23925001 |
| ORY TS513 | Orycteropus afer | Aardvark | Tubulidentata | Myrmecophagous | NA | NA | Wild | South Africa | Tswalu Kalahari reserve | SRR23924999 | SRR23924998 |
| ORY TS566 | Orycteropus afer | Aardvark | Tubulidentata | Myrmecophagous | NA | NA | Wild | South Africa | Tswalu Kalahari reserve | SRR23924997 | SRR23924996 |
| PAN TS471 | Smutsia temminckii | Ground pangolin | Pholidota | Myrmecophagous | NA | NA | Wild | South Africa | Tswalu Kalahari reserve | SRR23924995 | SRR23924994 |
| PAN TS475 | Smutsia temminckii | Ground pangolin | Pholidota | Myrmecophagous | NA | NA | Wild | South Africa | Tswalu Kalahari reserve | SRR23924993 | SRR23924992 |
| PAN TS482 | Smutsia temminckii | Ground pangolin | Pholidota | Myrmecophagous | NA | NA | Wild | South Africa | Tswalu Kalahari reserve | SRR23924991 | SRR23924990 |
| PAN TS488 | Smutsia temminckii | Ground pangolin | Pholidota | Myrmecophagous | NA | NA | Wild | South Africa | Tswalu Kalahari reserve | SRR23924988 | SRR23924987 |
| PAN TS489 | Smutsia temminckii | Ground pangolin | Pholidota | Myrmecophagous | NA | NA | Wild | South Africa | Tswalu Kalahari reserve | SRR23924986 | SRR23924985 |
| PAN TS525 | Smutsia temminckii | Ground pangolin | Pholidota | Myrmecophagous | NA | NA | Wild | South Africa | Tswalu Kalahari reserve | SRR23924984 | SRR23924983 |
| PRO TDR7 | Proteles cristatus | Southern aardwolf | Carnivora | Myrmecophagous | NA | NA | Wild | South Africa | Tussen-die-Riviere reserve | SRR23924982 | SRR23924981 |
| PRO TDR17 | Proteles cristatus | Southern aardwolf | Carnivora | Myrmecophagous | NA | NA | Wild | South Africa | Tussen-die-Riviere reserve | SRR23924980 | SRR23924979 |
| PRO TDR22 | Proteles cristatus | Southern aardwolf | Carnivora | Myrmecophagous | NA | NA | Wild | South Africa | Tussen-die-Riviere reserve | SRR23924977 | SRR23924976 |
| PRO TDR49 | Proteles cristatus | Southern aardwolf | Carnivora | Myrmecophagous | NA | NA | Wild | South Africa | Tussen-die-Riviere reserve | SRR23924975 | SRR23924974 |
| PRO TDR62 | Proteles cristatus | Southern aardwolf | Carnivora | Myrmecophagous | NA | NA | Wild | South Africa | Tussen-die-Riviere reserve | SRR23924973 | SRR23924972 |
| PRO TDR67 | Proteles cristatus | Southern aardwolf | Carnivora | Myrmecophagous | NA | NA | Wild | South Africa | Tussen-die-Riviere reserve | SRR23924971 | SRR23924970 |
| TAM M3075 | Tamandua tetradactyla | Southern tamandua | Pilosa | Myrmecophagous | Male | Adult | Wild | French Guiana | coll. JAGUARS | SRR23924969 | SRR23924968 |
| TAM M5331 | Tamandua tetradactyla | Southern tamandua | Pilosa | Myrmecophagous | Male | Adult | Wild | French Guiana | coll. JAGUARS | SRR23924966 | SRR23924965 |

**TABLE 1** Detailed sample information for the 33 fecal samples collected[a,b] (Continued)

| Sample name | Species name | Common name | Order | Diet | Sex | Age | Wild/ Captive | Country of origin | Location | Accession number |
|---|---|---|---|---|---|---|---|---|---|---|
| TAM M5584 | *Tamandua tetradactyla* | Southern tamandua | Pilosa | Myrmecophagous | Female | Adult | Wild | French Guiana | coll. JAGUARS | SRR23924964  SRR23924963 |

[a]Diet was determined based on field observations (i.e., dissections) and the literature (11, 74–78).
[b]Long- and short-reads are available on SRA under BioProject PRJNA942254 (https://www.ncbi.nlm.nih.gov/bioproject/PRJNA942254/).

### Short-read sequencing

Short-read shotgun metagenomic Illumina sequencing was done for all samples to generate data for long-read assemblies polishing. Either the first or second extraction was used for sequencing as they have previously been shown to be both suitable for short-read sequencing (79). Library preparation and Illumina sequencing on a Nova-Seq instrument were outsourced to Novogene Europe (Cambridge, UK) to generate metagenomes using 50 million of 150 bp paired-end reads each (15 Gb of raw data per sample) (Table S1A). Two samples of two *Dasypus* species (DASY M1746 and DASY VLD168) could not be sequenced using short reads due to low quantity of starting material and were not included in the following analyses; resulting in 31 samples being analyzed. Short-read metagenomic sequencing of soil samples was performed following the same protocol.

## Data filtering

### Long-read data

Sequencing adapters were removed with Porechop v0.2.4 (81) used with default parameters. Reads shorter than 200 bp were removed using Filtlong v0.2.1 (82). No quality filtering was performed at this stage as it was previously done during basecalling (Qscore >10).

Host mitogenomes were downloaded for the nine myrmecophagous species of our dataset from the National Center for Biotechnology Information (NCBI) Genbank database (Table S1A). Long-read metagenomes were mapped to the host mitogenome with Minimap2 v2.17 (83, 84) with the ONT preset (-*ax map-ont*) for Oxford Nanopore reads. This allowed us to confirm the host species. Host mitogenomes were then assembled from the mapping reads with Flye v2.8.3 (85) with default parameters for ONT raw reads (--*nano-raw*). Mapping of host mitogenomic reads on host mitogenomes was visually checked with Geneious Prime 2022.0.2 (86), and mitochondrial reads were removed from the metagenomes for downstream analyses.

The same was done to remove host nuclear reads by mapping long-read metagenomes against the host genome using Minimap2 v2.17 (83, 84) using the Nanopore read option (-*ax map-ont*) (Table S1A). The *Dasypus novemcinctus* RefSeq genome assembly (Dasnov 3.0; GCF_000208655.1) was downloaded from Genbank. The *Orycteropus afer* genome was downloaded from the DNA Zoo database (87) (HiC assembly based on the draft assembly of Di Palma et al., unpublished). *Myrmecophaga tridactyla*, *Tamandua tetradactyla*, *Smutsia gigantea,* and *Proteles cristatus* genomes have been previously reconstructed using long (ONT) and short (Illumina) reads assembled with MaSuRCA v3.2.9 (88) [see (89) for a detailed description of the hybrid assembly process]. For *Cabassous unicinctus*, we used the Discovar draft genome assembly generated by (90). For species lacking an available reference genome, the genome of the closest relative in our dataset was used: *Smutsia gigantea* for *S. temminckii* and *Dasypus novemcinctus* for *D. kappleri* and *Dasypus* sp. nov. FG. Host reads were removed from the metagenomes for downstream analyses. One southern naked-tailed armadillo sample (CAB M3141) and one giant anteater sample (MYR M5295) presented a high proportion of host reads (>95%) and were excluded from the following analyses. The final dataset thus included 29 samples.

Finally, contaminant human reads were removed following the same approach and using the telomere-to-telomere human genome assembly (GCA009914755.3) as reference (Table S1A).

### Short-read data

Illumina sequencing adapters were removed using FASTP v0.20.0 with default quality filtering parameters (91). Host and human reads were removed using the same approach as for long-read metagenomes but the mapping was done with bowtie2 v2.3.5 (92) with default parameters (Table S1A).

Samtools v1.7 (93) was used to manipulate long- and short-read mapping files.

## Metagenome assembly

Long-read metagenomes were assembled for each sample using metaFlye v2.9 (94) with default parameters for ONT reads basecalled with Guppy v5+ (*--nano-hq*, recommended mode when <5% of sequencing errors is expected) and the strain mode (*--keep-haplotypes*), which prevents closely related strains represented by different paths in the assembly graph to be collapsed. Long-read metagenome assemblies were then polished with short-reads using Pilon v1.24 (95) with default parameters. Short-read metagenomes were assembled using metaSPAdes v3.11.0 (96) and MEGAHIT v1.1.2 (97) with default parameters.

Assembly statistics were computed with anvi'o v7 (41) (Tables S1B and C). Anvi'o first generates a contig database in which *k*-mer frequencies for each contig are computed ($k = 4$). Prodigal v2.6.3 (98) is then used to predict archaeal and bacterial open reading frames and estimate the number of genes. Finally, microbial single-copy core genes (SCGs) are searched using HMMER (99) with the hmmscan v3.2.1 program (default in anvi'o).

## Reconstruction of bacterial genomes

### Genome binning and bins selection

Prior to binning, reads were mapped to metagenome assemblies to obtain the coverage information needed for binning. This was done using Bowtie2 v2.2.9 (92) for short reads and Minimap2 v2.17 (83, 84) for long reads. Genome binning was conducted on single long-read polished assemblies and short-read assemblies separately (trimmed to keep contigs longer than 1000 bp) using metaBAT2 v2.15 (100) with default parameters (minimum size of contigs set to 2,500 bp and bin minimal length set to 200 kb).

Completeness and redundancy of bins were estimated using anvi'o v7 (Tables S1D and E), which relies on the detection of SCGs. Bins with more than 90% completeness and less than 5% redundancy were selected for downstream analyses. Based on these criteria, 239 genome bins were selected from the long-read assemblies and 254 from the short-read assemblies. Genome bins statistics were computed using anvi'o v7 as done for the metagenome assemblies (Tables S1D and E).

### Genome bins dereplication

To remove potentially redundant bins, a dereplication step was performed on the set of selected bins using dRep v3.3.0 (101). dRep first filters genomes based on their length (>50 000 by default) and their completeness and redundancy as computed with CheckM v1.1.11 (102) (>75% completeness, <25% redundancy by default). dRep then clusters genomes based on an average nucleotide identity (ANI) threshold of 90% using the Mash algorithm v2.3 (103). Genomes having at least 90% ANI are then clustered using fastANI v1.33 (104), here with a secondary ANI threshold of 98% (recommended to avoid mis-mapping of metagenomic reads against bins, e.g., to study their distribution across metagenomes). By default, at least 10% of the genome is compared (minimum alignment fraction). Two bins did not pass dRep quality filtering in each dataset and were not included in the analysis. In the long-read dataset, 38 genomes were removed by dRep with an ANI of 98%, resulting in 201 unique genome bins. In the short-read dataset, 48 genomes were removed, resulting in 206 unique genome bins. Selected genome bin statistics were compared between bins reconstructed from long-read and short-read assemblies using anvi'o v7 (see supplementary results part 1 available via Zenodo).

To determine whether similar genome bins were reconstructed using long-read polished assemblies ($n = 201$) or short-read assemblies ($n = 206$), dRep was also run on the set of all selected bins ($n = 407$) with 98% ANI to remove redundant genome bins (93 genomes removed, resulting in 314 nonredundant high-quality bins). To capture the majority of the bacterial taxa present in our samples, we decided to combine

high-quality selected bins reconstructed from both long- and short-read assemblies dereplicated with an ANI of 98% in our final dataset for downstream analyses (*n* = 314; Table S1F) (see supplementary results part 1 and dRep output results available via Zenodo).

## Taxonomic assignment of selected bins

The taxonomy of selected genome bins reconstructed from long- and short-read assemblies was assessed using anvi'o v7, which uses 22 SCGs and the taxonomy of the genomes defined by the Genome Taxonomy Database release 7 (GTDB) (38) from which these genes have been extracted (Table S1F).

The 314 selected bins were placed in a phylogeny of 2566 reference prokaryotic genomes downloaded from Genbank using PhyloPhlAn v3.0.58 (42). PhyloPhlAn first searches universal prokaryotic marker genes (option *-d* phylophlan, a database comprising 400 markers) (105), here using Diamond v2.0.6 (106). Marker genes were then aligned using MAFFT v7.475 (107), and cleaned using trimAl v1.4 (108). Next, marker gene alignments were concatenated. Finally, the phylogeny was inferred using IQ-TREE v2.0.3 (109) under the LG model. PhyloPhlAn was run with the *–diversity* high and *–fast* options, which together set a range of parameters for the reconstruction of high-ranked taxonomic-level phylogenies (42). By default, at least 100 markers of the PhyloPhlAn database must be present in a genome for it to be included in the analysis (*--min_num_markers* 100) and each marker should be found in at least four genomes to be included in the analysis (*--min_num_entries* 4). Seventy of the reference genomes had less than 100 markers and were excluded from the analysis resulting in a total of 2810 genomes in the final reconstructed tree. The phylogeny was rooted with Archaea so the bacteria were monophyletic. Using the same PhyloPhlAn parameters, a phylogeny of the set of the 314 selected bacterial genome bins was also inferred (Fig. S1). This phylogeny was rooted according to (110), placing the root of the bacterial tree separating Terrabacteria (e.g., Firmicutes and Actinobacteriota) and Gracilicutes (e.g., Proteobacteria, Desulfobacterota, Campylobacterota, and Fibrobacteres-Chlorobi-Bacteroidetes group).

## Distribution of selected bins in gut metagenomes

Short-read metagenomes were mapped against the 314 selected bins using bowtie2 v2.3.5 with local sensitive alignment parameters. The distribution of bins across samples was explored using anvi'o v7. Anvi'o enables us to compute the percentage of mapped reads on each bin for each metagenome and computes statistics such as the coverage, percentage of recruited reads, abundance, and detection (horizontal coverage) of bins across metagenomes (samples). The distribution across samples was studied using the anvi'o interface by visualizing the detection of selected genome bins across samples. A detection threshold of 0.25% of the reference covered by at least one read was chosen to consider a bin detected in a sample. The number of bins shared among the different samples and host species and bins specific to a sample or host species were calculated. Applying the same approach, the distribution of selected bins reconstructed from the aardvark, ground pangolin, and southern aardwolf samples (*n* = 140) was explored in the eight soil samples collected in South Africa near feces sampling sites (Fig. S4).

## Identification of bacterial chitinase genes

Chitinase genes were searched in selected genome bins. More specifically, sequences of enzymes belonging to the glycoside hydrolase 18 (GH18) family (comprising chitinases and chitin-binding proteins) as determined by the classification of carbohydrate active enzymes (CAZymes) (31) were scanned using dbCAN2 (111) with default parameters. dbCAN2 is designed to annotate CAZymes sequences in genomes using an HMMER search (99) against the dbCAN HMM database containing HMM models for each CAZyme family. Proteins were predicted using Prodigal (98).

The amino acid sequences of 420 identified GH18 genes were imported in Geneious Prime 2022.0.2. Sequences were aligned using MAFFT v7.450 with default parameters. The alignment was cleaned by removing sites not present in at least 50% of the sequences, resulting in an alignment of 390 amino acids. The chitinase gene tree was inferred with RAxML v8.2.11 (112) under the LG + G model and the rapid hill-climbing algorithm for topology rearrangements. After removing 26 sequences that did not align correctly and had very long branches, the final alignment included 394 sequences. The phylogeny was rooted with the clades containing sequences without chitinolytic sites as they were divergent from the other sequences. The conserved chitinolytic site (DXXDXDXE) (48, 49), typical of chitin-degrading enzymes, was searched in the aligned sequences. BLAST searches against the NCBI nonredundant protein database were conducted to assess whether identified GH18 genes were similar to known microbial chitinases.

## ACKNOWLEDGMENTS

We are indebted to Lionel Hautier, Rémi Allio, Fabien Condamine, Sérgio Ferreira-Cardoso, Pierre-Henri Fabre, Quentin Martinez, Mathilde Barthe, Aude Caizergues, Nathalie Delsuc, Jessica Briner, Tshediso Putsane, Baptiste Chenet, Edith Guilloton, Solène Lefort, Margo Traimond, Michel Blanc, Bertrand Goguillon, and Antoine Baglan for their help with sampling. Fieldwork sessions at Tswalu Kalahari Reserve (South Africa) were conducted under the auspices of the Tswalu Foundation thanks to Duncan MacFadyen (Oppenheimer Generations Research and Conservation), Dylan Smith, and Gus van Dyk. We thank two anonymous reviewers for their helpful comments. Computational analyses benefited from the Montpellier Bioinformatics Biodiversity (MBB) platform.

This work has been supported by grants from the European Research Council (ConvergeAnt project: ERC-2015-CoG-683257) and Investissements d'Avenir of the Agence Nationale de la Recherche (CEBA: ANR‐10‐LABX‐25‐01; CEMEB: ANR‐10‐LABX‐0004). The JAGUARS collection, managed by Kwata NGO, is supported by the Collectivité Territoriale de Guyane and the Direction Générale des Territoires et de la Mer de Guyane.

This is contribution ISEM 2023-121-SUD of the Institut des Sciences de l'Evolution de Montpellier.

## AUTHOR AFFILIATIONS

[1]Institut des Sciences de l'Evolution de Montpellier (ISEM), Univ Montpellier, CNRS, IRD, Montpellier, France

[2]CIC AG/Inserm 1424, Centre Hospitalier de Cayenne Andrée Rosemon, Cayenne, French Guiana, France

[3]Tropical Biome and immunopathology, Université de Guyane, Labex CEBA, DFR Santé, Cayenne, French Guiana, France

[4]Brain Function Research Group, School of Physiology, University of the Witwatersrand, Johannesburg, South Africa

[5]Centre for African Ecology, School of Animals, Plant, and Environmental Sciences, University of the Witwatersrand, Johannesburg, South Africa

[6]Department of Biology, Valdosta State University, Valdosta, Georgia, USA

[7]National Museum and Centre for Environmental Management, University of the Free State, Bloemfontein, South Africa

[8]Institut Pasteur de la Guyane, Cayenne, French Guiana, France

[9]Kwata NGO, Cayenne, French Guiana, France

[10]Evolutionary Biology of the Microbial Cell, Institut Pasteur, Université Paris Cité, Paris, France

## AUTHOR ORCIDs

Sophie Teullet  http://orcid.org/0000-0003-2693-1797

Marie-Ka Tilak  http://orcid.org/0000-0001-8995-3462
Roxane Schaub  http://orcid.org/0000-0001-5742-3280
Nora M. Weyer  http://orcid.org/0000-0002-8753-9222
Wendy Panaino  http://orcid.org/0000-0002-8494-744X
Andrea Fuller  http://orcid.org/0000-0001-6370-8151
W. J. Loughry  http://orcid.org/0000-0001-8214-9893
Nico L. Avenant  http://orcid.org/0000-0002-5390-9010
Benoit de Thoisy  http://orcid.org/0000-0002-8420-5112
Guillaume Borrel  http://orcid.org/0000-0003-4893-8180
Frédéric Delsuc  http://orcid.org/0000-0002-6501-6287

## FUNDING

| Funder | Grant(s) | Author(s) |
| --- | --- | --- |
| EC \| European Research Council (ERC) | 683257 | Sophie Teullet |
| | | Amandine Magdeleine |
| | | Frédéric Delsuc |
| Agence Nationale de la Recherche (ANR) | ANR-10-LABX-0025 | Sophie Teullet |
| | | Marie-Ka Tilak |
| | | Amandine Magdeleine |
| | | Roxane Schaub |
| | | Benoit de Thoisy |
| | | Frédéric Delsuc |
| Agence Nationale de la Recherche (ANR) | ANR-10-LABX-0004 | Sophie Teullet |
| | | Marie-Ka Tilak |
| | | Amandine Magdeleine |
| | | Frédéric Delsuc |

## AUTHOR CONTRIBUTIONS

Sophie Teullet, Data curation, Formal analysis, Investigation, Visualization, Writing – original draft, Conceptualization | Marie-Ka Tilak, Investigation, Resources, Supervision, Writing – review and editing | Amandine Magdeleine, Investigation, Resources | Roxane Schaub, Resources, Writing – review and editing | Nora M. Weyer, Resources, Writing – review and editing | Wendy Panaino, Resources, Writing – review and editing | Andrea Fuller, Resources, Supervision, Writing – review and editing | W. J. Loughry, Resources, Writing – review and editing | Nico L. Avenant, Resources, Writing – review and editing | Benoit de Thoisy, Resources, Supervision, Writing – review and editing | Guillaume Borrel, Formal analysis, Writing – review and editing | Frédéric Delsuc, Conceptualization, Data curation, Formal analysis, Investigation, Visualization, Resources, Supervision, Writing – review and editing, Funding acquisition, Project administration

## DATA AVAILABILITY

Raw RNAseq short Illumina and long ONT reads have been submitted to the Short Read Archive (SRA) of the National Center for Biotechnology Information (NCBI) and are available under BioProject number PRJNA942254. Metagenome assemblies, genome bins (all and selected), phylogenetic datasets (alignments and corresponding trees), and other supplementary materials and results are available from Zenodo 7995394.

## ADDITIONAL FILES

The following material is available online.

## Supplemental Material

**Fig. S1 (mSystems00388-23-S0001.pdf).** Phylogeny of the 314 high-quality selected bins reconstructed from long- and short-read assemblies.

**Fig. S2 (mSystems00388-23-S0002.pdf).** Myrmecophagous-specific clades within Bacteroidetes and Proteobacteria.

**Fig. S3 (mSystems00388-23-S0003.svg).** Bacterial families of host-species-specific and shared genome bins carrying or not carrying GH18 genes.

**Fig. S4 (mSystems00388-23-S0004.svg).** Detection of the 140 high-quality selected bins reconstructed from aardvark, ground pangolin, and southern aardwolf samples.

**Table S1 (mSystems00388-23-S0005.xlsx).** Raw results of the different analyses conducted on each gut metagenome to reconstruct high-quality genome bins from raw metagenomic data for each dataset (long- and short-reads).

**Table S2 (mSystems00388-23-S0006.xlsx).** Presence/absence of the 314 high-quality selected genome bins across the 29 gut metagenomes of the nine focal myrmecophagous species.

**Table S3 (mSystems00388-23-S0007.pdf).** Proportion of chitinolytic genome bins (genomes having at least one GH18 with an active chitinolytic site) detected in the nine focal myrmecophagous species.

**Table S4 (mSystems00388-23-S0008.pdf).** Detailed sample information for the eight soil samples collected in South Africa.

## Open Peer Review

**PEER REVIEW HISTORY (review-history.pdf).** An accounting of the reviewer comments and feedback.

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
