## [Reviewer comments · mSystems]

Metagenomics uncovers dietary adaptations for chitin digestion in the gut microbiota of convergent myrmecophagous mammals

Sophie Teullet, Marie-Ka Tilak, Amandine Magdeleine, Roxane Schaub, Nora Weyer, Wendy Panaino, Andrea Fuller, William Loughry, Nico Avenant, Benoit de Thoisy, Guillaume Borrel, and Frederic Delsuc

Corresponding Author(s): Sophie Teullet, Institut des Sciences de l'Evolution de Montpellier

Review Timeline:

Submission Date:	April 20, 2023
Editorial Decision:	May 17, 2023
Revision Received:	June 8, 2023
Accepted:	June 19, 2023

Editor: Sarah Hird

Reviewer(s): Disclosure of reviewer identity is with reference to reviewer comments included in decision letter(s). The following individuals involved in review of your submission have agreed to reveal their identity: Connie Rojas (Reviewer #1)

Transaction Report:

DOI: <https://doi.org/10.1128/msystems.00388-23>

Hello Dr. Teullet -

The reviewers and I all thought highly of this manuscript and it is well suited for mSystems. Please be sure to address all the comments of the reviewers before resubmitting. I especially echo Reviewer 1's comments about the figures. Figures 3 and 4 are not legible.

Thank you (standard form letter with additional information below) -
Sarah

May 17, 2023

Dr. Sophie Teullet
Institut des Sciences de l'Evolution de Montpellier
Montpellier
France

Re: mSystems00388-23 (**Metagenomics uncovers dietary adaptations for chitin digestion in the gut microbiota of convergent myrmecophagous mammals**)

Dear Dr. Sophie Teullet:

Thank you for submitting your manuscript to mSystems. We have completed our review and I am pleased to inform you that, in principle, we expect to accept it for publication in mSystems. However, acceptance will not be final until you have adequately addressed the reviewer comments.

Preparing Revision Guidelines

Please return the manuscript within 60 days; if you cannot complete the modification within this time period, please contact me. If you do not wish to modify the manuscript and prefer to submit it to another journal, please notify me of your decision immediately so that the manuscript may be formally withdrawn from consideration by mSystems.

Sincerely,

Sarah Hird

Editor, mSystems

Journals Department
Reviewer comments:

Reviewer #1 (Comments for the Author):

Great study! The manuscript was well-written and the findings were interesting and robust! My mainly minor comments focus on including more genus-level information to the results section, adding missing details to the methods section, and making the figures a bit more accessible to readers. My specific comments are in the attached file.

Reviewer #2 (Comments for the Author):

Teullet et al. report chitin degraders in myrmecophagous and insectivorous using nanopore and shotgun sequencing approaches. Overall the manuscript is well written. I have minor comments for the authors.

Keywords: the authors might want to consider adding microbiome or microbial symbionts or gut microbiota; genome binning isn't necessary instead the authors might want to consider adding genome assembly

Introduction:

The sentence with the term "non-flying mammals" can be better explained by giving some context [1 line]. For eg. the authors might want to add that in comparison to bats, phylosymbiosis is well explained for

Please rephrase this sentence "More specifically, symbiotic chitin-degrading bacteria have been identified, as well as bacterial chitinase genes, in the Malayan pangolin and the giant anteater.."

Please rephrase "Chitinolytic microbes might thus have been independently recruited to ensure chitin digestion, but not necessarily similar ones."

A part of the introduction focuses on convergent evolution - is this important here? The analyses do not necessarily focus on convergent evolution though. This is upto the authors to decide whether convergent evolution explanation is really necessary here. Rather a brief summary of chitinolytic microbes in mammals as well as the enzymes GH can be provided in the introduction [also because the authors have already provided the relevant information on divergent evolution in the discussion section].

Reviewer Comments mSystems00388-23

The study titled “Metagenomics uncovers dietary adaptations for chitin digestion in the gut microbiota of convergent myrmecophagous mammals” investigates the roles of gut microbes in host digestion of chitinous exoskeletons. They do this by reconstructing and functionally annotating genome bins from fecal samples of 9 sp. of myrmecophagous mammals. It is clear that the authors put a lot of time and effort into this study and the manuscript, which is very well-written. They present highly interesting findings, a detailed and transparent methods section, an informative discussion section, and all of the necessary supplementary material. Most of my comments, which I share below, are minor and focused on adding valuable details to the results or methods section.

**The manuscript did not contain line numbers (or page numbers) which makes it hard for reviewers to pinpoint exact sentences or areas in the manuscript. I would advise the authors include line numbers in any manuscript they plan on submitting to a journal.*

Introduction

Not sure the first paragraph of the introduction is necessary to include, given that it does not provide critical information necessary for understanding this study. I would suggest the authors open the introduction by talking about myrmecophagy (how widespread it is, and what is known about the microbiome taxonomic composition and function).

I would advise the authors highlight the motivation for this study more throughout the introduction (to examine whether the (microbiome/microbial) mechanisms of chitin exoskeleton digestion are convergent across the 9 species (and 5 orders) of myrmecophagous mammals). It is a large gap in the literature that is being addressed!

Not sure if the 2-3 sentences explaining what metagenomics are and why they are valuable are necessary to include in the manuscript given that this is widely known.

Methods

In this section, it is mentioned that two samples contained a disproportionate amount of host reads and were removed from the dataset, which brings samples from 33 to 31. But throughout the paper, 29 gut metagenomes are mentioned. What happened to the other two samples?

I appreciate the extra steps to remove both host and human contamination from the gut metagenomes!

In the ‘Genome binning and bins selection’ Table S11 is mentioned but I believe this should be Table S1?

Prior to annotating genome bins with dbCAN against CAZy/GH18, did the authors need to predict open-reading frames in the bins? I am not sure this is mentioned in the methods section.

Which GTDB release was used for taxonomic assignment?

How did the authors root the phylogenies they present in the main figures of the manuscript?

Results

The first sentence states, "Combining long-read and short-read assemblies, we were able to reconstruct 314 high-quality genome bins (156 and 158 from each dataset respectively)."

Can the authors add 'dereplicated' before 'high-quality'?

It would be helpful if in the first section of results the authors mention the % of the 314 bins that were assigned to Species, Genus, and Family levels. This will make it clear to the reader that many bins did not receive beyond Family level annotation and why authors are not mentioning specific bacterial genera when talking about the bins.

Can the authors mention a few of the names of the enzymes that form part of glycoside hydrolase family 18 (e.g. Chitinase domain-containing protein 1)? As a biologist I wanted to know a bit more about this protein family since it is the focus of this manuscript.

Throughout the results section, the authors mention a few of the bacterial families that had bins with GH18 sequences or active chitinolytic sites, but often do not mention any specific genera. Can the authors include a few of the bacterial genera in parentheses as examples? For the bins that did not receive a Genus or Species-level taxonomic assignment, maybe the authors can mention what their closest relatives were in GTDB? As a reader I wanted a bit more specificity that was provided. A bit more biology to the results narrative.

Super cool results, especially that some bins were in their own clade that did not include any GTDB genomes and that GH18 sequences were recovered in bins from across all 9 myrmecophagous orders!

I was interested in findings regarding the bin relative abundances! Which bins were the most abundant in the dataset, or in particular host species? What was their taxonomic classification?

Figures

Great figures! I know this may be difficult but if there is any way to make the bin IDs a bigger font size that would be valuable for readers. I found it almost impossible to read the bin IDs from the figures even after zooming in.

Is there a reason why the phylogenies in Figure 2 do not include the bin's taxonomic assignment alongside their bin ID or in lieu of their bin ID? Taxonomic classifications are more informative than random IDs. The same can be said for Figure 3.

Can the authors provide a table that lists all of the bin IDs, their taxonomic assignment, a column indicating whether they contained at least one GH18, whether they had an active chitinolytic site, and columns indicating their presence/absence across the 29 gut metagenomes? Sort of what Figure 3 or 4 show but in a tabular form. As a reader, I really wanted to see the taxonomic classifications of these bins. Table S1F sort of gets at this but not completely.

Discussion

Great discussion and interpretation of findings!

Data Availability

Can the authors include the https link to their Zenodo project in addition to the DOI? The DOI 10.5281/zenodo.7808597 takes me to a google search but <https://doi.org/10.5281/zenodo.7808597> takes me to their datasets

The Zenodo supplementary material was a bit disorganized and it was not clear which file included which supplementary figures or tables. It would be valuable for authors to reorganize or rename some of their files so it is more intuitive which files need to be downloaded depending on what one is looking for.

Editor comments:

E1.1: Thank you for submitting your manuscript to mSystems. We have completed our review and I am pleased to inform you that, in principle, we expect to accept it for publication in mSystems. However, acceptance will not be final until you have adequately addressed the reviewer comments.

AE1.1: Thank you for your efficient and positive assessment of our work. As detailed below, we have carefully addressed all reviewer comments.

Reviewer comments:

Reviewer #1:

R1.1: Great study! The manuscript was well-written and the findings were interesting and robust! My mainly minor comments focus on including more genus-level information to the results section, adding missing details to the methods section, and making the figures a bit more accessible to readers. My specific comments are in the attached file.

AR1.1: Thank you for your positive and constructive comments on our manuscript. We have taken into account your detailed suggestions in the revised version.

R1.2: The study titled “Metagenomics uncovers dietary adaptations for chitin digestion in the gut microbiota of convergent myrmecophagous mammals” investigates the roles of gut microbes in host digestion of chitinous exoskeletons. They do this by reconstructing and functionally annotating genome bins from fecal samples of 9 sp. of myrmecophagous mammals. It is clear that the authors put a lot of time and effort into this study and the manuscript, which is very well-written. They present highly interesting findings, a detailed and transparent methods section, an informative discussion section, and all of the necessary supplementary material. Most of my comments, which I share below, are minor and focused on adding valuable details to the results or methods section.

*The manuscript did not contain line numbers (or page numbers) which makes it hard for reviewers to pinpoint exact sentences or areas in the manuscript. I would advise the authors include line numbers in any manuscript they plan on submitting to a journal.

AR1.2: Sorry for the lack of line numbering. We added it to the revised version.

R1.3: Introduction. Not sure the first paragraph of the introduction is necessary to include, given that it does not providing critical information necessary for understanding this study. I would suggest the authors open the introduction by talking about myrmecophagy (how widespread it is, and what is known about the microbiome taxonomic composition and function).

AR1.3: We reduced this first paragraph to introduce our myrmecophagy model system earlier. We also removed the part about the morphological adaptations of myrmecophagous species and instead detailed a bit more in the second paragraph the previous results from Delsuc *et*

a/ (2014) on the taxonomic composition of their gut microbiota (see Marked copy of the manuscript).

R1.4: I would advise the authors highlight the motivation for this study more throughout the introduction (to examine whether the (microbiome/microbial) mechanisms of chitin exoskeleton digestion are convergent across the 9 species (and 5 orders) of myrmecophagous mammals). It is large gap in the literature that is being addressed!

AR1.4: It is indeed the focus of our study and we emphasized it more in the revised introduction (see Marked copy of the manuscript).

R1.5: Not sure if the 2-3 sentences explaining what metagenomics are and why they are valuable are necessary to include in the manuscript given that this is widely known.

AR1.5: Thanks for this comment. Indeed, it was not really relevant to introduce our study. We removed this part in the introduction of the revised manuscript (see Marked copy of the manuscript).

R1.6: Methods. In this section, it is mentioned that two samples contained a disproportionate amount of host reads and were removed from the dataset, which brings samples from 33 to 31. But throughout the paper, 29 gut metagenomes are mentioned. What happened to the other two samples?

AR1.6: Sorry if it was not clear. From the 33 samples we had, two were not included in the analyses because we did not have enough remaining material to sequence them using Illumina short reads. They were thus only sequenced with Nanopore long reads. This was mentioned in part 3.2 of the Material and Methods section but we re-emphasized it in the revised version to make it hopefully clearer (see Marked copy of the manuscript). Two more samples were also removed due to their high amount of host reads. In total, four samples were therefore discarded from our initial dataset.

R1.7: I appreciate the extra steps to remove both host and human contamination from the gut metagenomes!

AR1.7: Thanks. It was necessary since some samples contained a very large proportion of host reads. As explained above, two individuals unfortunately had to be excluded from further analyses because of this filter.

R1.8: In the 'Genome binning and bins selection' Table S11 is mentioned but I believe this should be Table S1?

AR1.8: Indeed, it was a typo. Table S1 is now correctly cited in the revised version.

R1.9: Prior to annotating genome bins with dbCAN against CAZy/GH18, did the authors need to predict open-reading frames in the bins? I am not sure this is mentioned in the methods section.

AR1.9: The prediction of ORFs in genome bins was done within *anvi'o* v7 with Prodigal v2.6.3 (see Material and Methods part 5). When the input of dbCAN2 consists of bacterial genome bins, Prodigal is used to predict protein sequences before doing the HMM search to identify CAZymes families.

R1.10: Which GTDB release was used for taxonomic assignment?

AR1.10: GTDB release 7 was used for the taxonomic assignment of the genome bins. This is now indicated in the Material and Methods and we changed the associated reference accordingly.

R1.11: How did the authors root the phylogenies they present in the main figures of the Manuscript?

AR1.11: The phylogeny of the 314 high-quality selected bins (used in Figures 3 and 4 and presented in Figure S1) was rooted according to Coleman *et al* (2021). This is now indicated in the Material and Methods section. Figure 1 was rooted on Archaea so that Bacteria would be monophyletic even if it is now recognized that Eukaryota are part of the Archaea. Figure 2 was rooted on the clade including sequences without active site and that are similar to LysM and SH3 domain-containing proteins.

R1.12: Results. The first sentence states, "Combining long-read and short-read assemblies, we were able to reconstruct 314 high-quality genome bins (156 and 158 from each dataset respectively)." Can the authors add 'dereplicated' before 'high-quality'?

AR1.12: It has been added in the revised version.

R1.13: It would be helpful if in the first section of results the authors mention the % of the 314 bins that were assigned to Species, Genus, and Family levels. This will make it clear to the reader that many bins did not receive beyond Family level annotation and why authors are not mentioning specific bacterial genera when talking about the bins.

AR1.13: We added a sentence in the first paragraph of the Results section to detail these percentages (see Marked copy of the manuscript).

R1.14: Can the authors mention a few of the names of the enzymes that form part of glycoside hydrolase family 18 (e.g. Chitinase domain-containing protein 1)? As a biologist I wanted to know a bit more about this protein family since it is the focus of this manuscript.

AR1.14: Thank you for this comment, it was indeed important to better introduce the enzyme family we work on. Microbial chitinases and the GH18 family are now described in more detail in the Introduction of the revised version of the manuscript (see Marked copy of the manuscript).

R1.15: Throughout the results section, the authors mention a few of the bacterial families that had bins with GH18 sequences or active chitinolytic sites, but often do not mention any specific genera. Can the authors include a few of the bacterial genera in parentheses as examples? For the bins that did not receive a Genus or Species-level taxonomic assignment, maybe the

authors can mention what their closest relatives were in GTDB? As a reader I wanted a bit more specificity that was provided. A bit more biology to the results narrative.

AR1.15: Main bacterial genera included in the families mentioned in the results section have been added in the revised version (see Marked copy of the manuscript). Those without taxonomic assignment were not detailed but their taxonomically-assigned closest relatives can be seen in the phylogeny of the 314 selected genome bins (Fig S1).

R1.16: Super cool results, especially that some bins were in their own clade that did not include any GTDB genomes and that GH18 sequences were recovered in bins from across all 9 myrmecophagous orders!

AR1.16: Thank you! These are indeed very interesting and exciting results!

R1.17: I was interested in findings regarding the bin relative abundances! Which bins were the most abundant in the dataset, or in particular host species? What was their taxonomic classification?

AR1.17: We chose not to focus on abundances in this study because it was a bit difficult to make quantitative comparisons. For example, as we did not have the same amount of data for each sample using long reads, there might be sequencing biases resulting in certain genome bins being highly covered. Moreover, our aim was mainly to identify chitinolytic Bacteria in order to highlight the potential role of the gut microbiome in chitin digestion. This is the reason why we briefly discuss the results of abundance estimations only for bins shared between all or most host orders that are carrying GH18 genes.

More specifically:

- The genome bin with the highest maximal abundance in a sample (i.e. 198; for a mean abundance across samples of 9.8) belongs to the genus *Fusobacterium* (Fusobacteriaceae), was detected in three host species, and does not carry GH18 genes.
- Another genome bin with a maximal abundance of 110 (and a mean abundance across samples of 9.4) also belongs to this genus, was detected in two host species, and does not carry any GH18 gene.
- Another bin with a maximal abundance of 100 (and a mean abundance across samples of 6.9) belongs to *Bacteroidetes fragilis*, was detected in six host species, and does not carry any GH18 gene.
- A bin belonging to Enterobacteriaceae is detected across samples with a mean abundance of 12.4 and a maximal abundance of 129. It was detected in six host species and it does not carry any GH18 gene.
- A bin assigned to *Pseudomonas formosensis* was detected across samples with a mean abundance of 6.9 and a maximal abundance of 170. It was detected in one host species and does not carry any GH18 gene.

Bins with the highest abundances thus did not carry any GH18 gene and were therefore not of special interest for our discussion of chitin-digestion potential in myrmecophagous species.

R1.18: Figures. Great figures! I know this may be difficult but if there is any way to make the bin IDs a bigger font size that would be valuable for readers. I found it almost impossible to read the bin IDs from the figures even after zooming in.

AR1.18: Thank you! We tried to correct this as much as possible in the revised versions of our figures.

R1.19: Is there a reason why the phylogenies in Figure 2 do not include the bin's taxonomic assignment alongside their bin ID or in lieu of their bin ID? Taxonomic classifications are more informative than random IDs. The same can be said for Figure 3.

AR1.19: We added the information on the genus to which the bins were assigned to, as a prefix to bins' names in the main figures of the revised version.

R1.20: Can the authors provide a table that lists all of the bin IDs, their taxonomic assignment, a column indicating whether they contained at least one GH18, whether they had an active chitinolytic site, and columns indicating their presence/absence across the 29 gut metagenomes? Sort of what Figure 3 or 4 show but in a tabular form. As a reader, I really wanted to see the taxonomic classifications of these bins. Table S1F sort of gets at this but not completely.

AR1.20: Thank you for this suggestion. We now provide such a summary table, which corresponds to Table S2 in the revised version.

R1.21: Discussion. Great discussion and interpretation of findings!

AR1.21: Thank you!

R1.22: Data Availability. Can the authors include the https link to their Zenodo project in addition to the DOI? The DOI 10.5281/zenodo.7808597 takes me to a google search but <https://doi.org/10.5281/zenodo.7808597> takes me to their datasets

AR1.22: Sorry for this, it has been corrected in the revised version and the link to the updated Zenodo v2 has been added.

R1.23: The Zenodo supplementary material was a bit disorganized and it was not clear which file included which supplementary figures or tables. It would be valuable for authors to reorganize or rename some of their files so it is more intuitive which files need to download depending on what one is looking for.

AR1.23: We reorganized the updated Zenodo (v2) by providing the corresponding datasets for each figure. Supplementary figures and corresponding datasets were also included as well as metagenomes and genomes bins, as they were in the first version.

Reviewer #2:

R2.1: Teullet et al. report chitin degraders in myrmecophagous and insectivorous using nanopore and shotgun sequencing approaches. Overall the manuscript is well written. I have minor comments for the authors.

AR2.1: Thank you for your positive assessment and comments.

R2.2: Keywords: the authors might want to consider adding microbiome or microbial symbionts or gut microbiota; genome binning isn't necessary instead the authors might want to consider adding genome assembly

AR2.2: This has been done in the revised version in which we added "gut microbiota" and replaced "genome binning" by "genome assembly".

R2.3: Introduction. The sentence with the term "non-flying mammals" can be better explained by giving some context [1 line]. For eg. the authors might want to add that in comparison to bats, phylosymbiosis is well explained for

AR2.3: When modifying the introduction according to reviewer 1 comments (see AR1.4), this sentence has been removed as we do not really discuss phylosymbiosis (see Marked version of the manuscript).

R2.4: Please rephrase this sentence "More specifically, symbiotic chitin-degrading bacteria have been identified, as well as bacterial chitinase genes, in the Malayan pangolin and the giant anteater.."

AR2.4: This sentence has been modified in the revised version (see Marked version of the manuscript).

R2.5: Please rephrase "Chitinolytic microbes might thus have been independently recruited to ensure chitin digestion, but not necessarily similar ones."

AR2.5: This sentence has been rephrased in the revised version (see Marked version of the manuscript).

R2.6: A part of the introduction focuses on convergent evolution - is this important here? The analyses do not necessarily focus on convergent evolution though. This is up to the authors to decide whether convergent evolution explanation is really necessary here.

AR2.6: Indeed, it is not the focus of our study, we removed that part in the revised version and instead focused more on the taxonomic composition of the gut microbiota of myrmecophagous mammals as suggested by reviewer 1 (see AR1.4 and Marked version of the manuscript).

R2.7: Rather a brief summary of chitinolytic microbes in mammals as well as the enzymes GH can be provided in the introduction [also because the authors have already provided the relevant information on divergent evolution in the discussion section].

AR2.7: Thank you for this suggestion. We detailed these two points in the revised version (see Marked version of the manuscript).

June 19, 2023

Dr. Sophie Teullet
Institut des Sciences de l'Evolution de Montpellier
Montpellier
France

Re: mSystems00388-23R1 (**Metagenomics uncovers dietary adaptations for chitin digestion in the gut microbiota of convergent myrmecophagous mammals**)

Dear Dr. Sophie Teullet:

Your manuscript has been accepted, and I am forwarding it to the ASM Journals Department for publication. For your reference, ASM Journals' address is given below. Before it can be scheduled for publication, your manuscript will be checked by the mSystems production staff to make sure that all elements meet the technical requirements for publication. They will contact you if anything needs to be revised before copyediting and production can begin. Otherwise, you will be notified when your proofs are ready to be viewed.

If you would like to submit a potential Featured Image, please email a file and a short legend to msystems@asmusa.org. Please note that we can only consider images that (i) the authors created or own and (ii) have not been previously published. By submitting, you agree that the image can be used under the same terms as the published article. File requirements: square dimensions (4" x 4"), 300 dpi resolution, RGB colorspace, TIF file format.

We recognize that the video files can become quite large, and so to avoid quality loss ASM suggests sending the video file via <https://www.wetransfer.com/>. When you have a final version of the video and the still ready to share, please send it to mSystems staff at msystems@asmusa.org.

Sincerely,

Sarah Hird
Editor, mSystems

Journals Department
E-mail: mSystems@asmusa.org